# A New Method for Moving-Target HRRP via Double Step Frequency Verified by Simulation

**DOI:** 10.3390/s22239191

**Published:** 2022-11-26

**Authors:** Xiaofeng Shen, Zhihong Zhuang, Hongbo Wang, Feng Shu

**Affiliations:** School of Electronic and Optical Engineering, Nanjing University of Science and Technology, Nanjing 210094, China

**Keywords:** double-stepped frequency, high-range-resolution profile, phase cancellation

## Abstract

The stepped-frequency (SF) waveform is highly sensitive to the target motion, which induces range shift and echo spread in a high-range-resolution profile (HRRP). This paper proposes a method based on a cross-transmitted double-stepped frequency (DSF) waveform and the phase-cancellation technique. The proposed method obtains the stationary HRRP of the moving targets according to the inverse discrete Fourier transform (IDFT) and complex multiplication. The results also show that the proposed method eliminates the generated false peaks from the existing methods. As a result, the obtained signal-to-noise ratio (SNR) of the HRRP using the proposed method is improved. Due to the cross-transmitted DSF waveform application, the proposed method adapts to higher speed targets. The analysis and simulation results validate the effectiveness of the proposed approach.

## 1. Introduction

Generally, the range resolution depends on the bandwidth of a radar. Owing to hardware limitations, it is difficult to directly generate real-time large-bandwidth signals. There are two techniques used for obtaining large bandwidth signals, including the de-chirping technique (DT) [1] and the synthetic bandwidth technique (SBT). The DT effectively reduces the signal bandwidth before sampling when the swath is narrow. However, for a wide swath, it is invalid. Moreover, the SBT divides the broadband signal into several narrow-band signals for transmitting and receiving to synthesize the whole bandwidth based on signal processing techniques after sampling all the narrow-band signals. The stepped-frequency (SF) waveform [2] is a typical example of a waveform that applies to SBT. In fact, it transmits a series of narrow-band pulses, which are stepped in frequency from pulse to pulse and form a burst that covers a wide bandwidth. The high range resolution profile (HRRP) is obtained by applying inverse discrete Fourier transform (IDFT) to the echoes of the SF waveform. Due to their ability in project realization and high-range-resolution performance, the SF waveforms are widely used in remote sensing, such as synthetic aperture radar (SAR) [3], inverse SAR [4], ground penetrating radar [5], through-wall imaging radar [6], and weather radar [7].

It is demonstrated that the SF radar can realize target detection, tracking, imaging, and recongnition [8]. However, the SF waveforms are highly sensitive to the target’s motion. The target’s radial velocity generates two motion-induced phase terms in echoes, linear phase term (LPT), and quadratic phase term (QPT). These phase terms lead to range shift and echo spread in HRRP, respectively [9]. Therefore, target motion requires compensation before generating the HRRP in the SF radar system. There are two main categories of motion compensation in the SF waveform, i.e., the parametric and non-parametric methods.

In parametric methods, the HRRP of the moving-target is generated after performing precise velocity compensation for echoes. Therefore, velocity estimation forms the base of parametric methods. As mentioned above, the QPT caused by the target’s velocity leads to echo spread in HRRP. The degree of echo spread can be represented by waveform entropy [10]. The higher the target speed, the worse the echo spread in HRRP, and the value of the waveform entropy is bigger. If the QPT is compensated for by the actual velocity, the range profile is perfectly focused, and then the waveform entropy is minimized. Considering the waveform entropy of the range-profile after speed compensation as an objective function, the process of searching the waveform with minimum entropy can be used to estimate the target velocity. Similarly, minimum image contrast [11,12], nonlinear least squares [13], and maximum likelihood [14] can also be applied as objective functions to the velocity estimation. The aforementioned methods search for the optimal velocity based on an iterative process. Therefore, the good performance of these methods is obtained at the cost of higher computational complexity. In [15], a DSF waveform comprising positive and negative SF waveforms is designed. By multiplying the echoes between the adjacent coherent pulse intervals (CPIs) and applying IDFT, the target velocity is estimated. However, it is noteworthy that the cross-phase terms introduced by the extended target containing multiple scattering points are not considered in this method. These cross-phase terms reduce the signal-to-noise ratio (SNR) of HRRP. As a result, this method is not suitable for extended targets. The phase-derived velocity measurement (PDVM) is presented in [16,17]. The velocity of a target can be estimated by combining PDVM and wide envelope velocity measurement (WEVM) via the adjacent CPIs. However, the highly accurate results require noncoherent accumulation in multiple CPIs based on a smoothing filter, which makes the process time-consuming.

The non-parametric methods are based on the phase-cancellation technique (PCT). The parameters of the SF waveform in these methods are elaborately designed in advance. Complex multiplication can partially or completely eliminate the motion-induced phase terms, and then the HRRP can be obtained directly by applying IDFT. In [18], a new DSF waveform is designed containing two sequentially transmitted SF waveforms. Each SF waveform’s pulse repetition time (PRT) varies according to the pulse serial number. The QPT can be effectively eliminated based on the PCT presented in [18]. However, it is not possible to eliminate LPT. Another DSF waveform presented in [19] contains two sequentially transmitted SF waveforms. The PRT of the second SF waveform is twice that of the first waveform. The other parameters of these two waveforms are the same. The QPT and LPT can be eliminated simultaneously via a similar PCT. According to [19], since this method only uses IDFT and complex multiplication, it has low computational complexity. In addition, it is easy to generate this DSF waveform. However, due to the intrinsic property of this sequentially transmitted DSF waveform, the method presented in [19] is not suitable for high-speed moving targets. Moreover, false peaks appear in HRRP with the SNR reduction using the presented PCT in [18,19].

Considering the aforementioned factors, this work proposes a novel method for generating the stationary HRRP of high-speed moving targets by applying a new PCT based on the cross-transmitted DSF waveform. The main contributions of this research are summarized below:A cross-transmitted DSF waveform consisting of two SF waveforms is designed in this paper. Compared to the sequentially transmitted DSF waveform in [19], since the pulses in these two SF waveforms are transmitted alternately, this cross-transmitted DSF waveform is more suitable for high-speed targets than the sequentially transmitted DSF waveform. Additionally, the reason is analyzed in detail;A new PCT is also presented, which realizes the elimination of LPT and QPT simultaneously. Compared to the existing PCT in [19], the new PCT performs better under low SNR conditions. Moreover, there is no false peak generated by using the new PCT.

## 2. Influence of Target’s Velocity in HRRP

The normal SF waveform is shown in Figure 1. The transmitted signal of this SF waveform radar is mathematically expressed as:(1)$$s\left(t\right)=\sum_{n=0}^{N-1}rect\left(\frac{t-nT_{r}-\frac{T_{p}}{2}}{T_{p}}\right)e^{j(2\pi f_{n}t+\theta_{n})}$$
where *N* denotes the number of pulses in a burst, $T_{p}$ represents the pulse width, $T_{r}$ indicates the PRT, $f_{n}=f_{0}+n\Delta f$ denotes the transmitting frequency of the *n*th pulse, $f_{0}$ expresses the initial carrier frequency, Δf represents the frequency step, and $\theta_{n}$ is the initial phase. Assuming that one extended target has *Q* scattering points, the *n*th baseband signal of a burst after frequency mixing can be expressed as follows:(2)$$r_{n}\left(t\right)=\sum_{q=0}^{Q-1}a_{q}rect\left(\frac{t-nT_{r}-\frac{T_{p}}{2}-\tau_{q}\left(t\right)}{T_{p}}\right)e^{-j2\pi f_{n}\tau_{q}\left(t\right)}$$
where $a_{q}$ represents the echo amplitude of the *q*th scattering point. The initial range between the *q*th scattering point and the radar is $R_{q}$ and the target’s velocity is $v_{0}$. The round-trip delay of the *q*th scattering point is mathematically presented as follows:(3)$$\tau_{q}\left(t\right)=\frac{2(R_{q}-v_{0}t)}{c}$$
where *c* denotes the speed of light. This paper uses oversampling and the iso-range local-maximum pick-up algorithm [20] to reduce the sampling loss and promote echo SNR. Assuming that $t_{s}$ denotes the sampling period, $M=round\left(\frac{T_{r}}{t_{s}}\right)$ denotes the sampling number of one pulse. Suppose that the target does not move to other sampling cells during the repetition interval of the SF waveform and the target is in the *m*th sampling cell, the following echo signal is derived by substituting (Equation 3) into (Equation 2):(4)$$r\left(n,m\right)=\sum_{q=0}^{Q-1}a_{q}e^{j(\varphi_{0}+\varphi_{1})}e^{j[\varphi_{R_{q}}+\left(\varphi_{m}+\varphi_{L}\right)+\varphi_{Q}]}$$
where $\varphi_{0}=-2\pi f_{0}\frac{2R_{q}}{c}$, $\varphi_{1}=2\pi f_{0}\frac{2v_{0}mt_{s}}{c}$, $\varphi_{R_{q}}=-2\pi \Delta f\frac{2R_{q}}{c}n$, $\varphi_{m}=2\pi \Delta f\frac{2v_{0}mt_{s}}{c}n$, $\varphi_{L}=2\pi f_{0}\frac{2v_{0}T_{r}}{c}n$, $\varphi_{Q}=2\pi \Delta f\frac{2v_{0}T_{r}}{c}n^{2}$.

If $v_{0}=0$, applying IDFT to (Equation 4), the HRRP of the target can be obtained.
(5)$$R\left(k,m\right)=\frac{1}{N}\sum_{n=0}^{N-1}r\left(n,m\right)e^{j\frac{2\pi }{N}kn}\approx \sum_{q=0}^{Q-1}a_{q}e^{j(\varphi_{0}+\varphi_{k})}sinc\left(k-k_{q}\right)$$
where $\varphi_{k}=\pi \frac{N-1}{N}\left(k-k_{q}\right)$, $k_{q}=\frac{R_{q}}{\Delta r}$, B=NΔf expresses the synthetic bandwidth, and $\Delta r=\frac{c}{2B}$ denotes the range resolution cell of HRRP. $\left(\varphi_{0}+\varphi_{k}\right)$ does not influence the amplitude of HRRP. Considering $N_{u}=floor\left(\frac{k_{q}}{N}\right)$, where $k=k_{q}-N_{u}N$, |R(k,m)| attains the maximum value. Therefore, the range of the *q*th scattering point is calculated as follows:(6)$$R_{q}=\left(k+N_{u}N\right)\Delta r$$
where $N_{u}$ is determined by the position of the sampling cell.

To facilitate the illustration of the influence of target velocity in HRRP, (Equation 4) is rewritten as:(7)$$r\left(n,m\right)=\sum_{q=0}^{Q-1}a_{q}e^{j(\varphi_{0}+\varphi_{1})}e^{j(2\pi f_{0}{}^{\prime }T_{s}+\pi \mu T_{s}^{2})}$$
(8)$$\left\{\begin{array}{c} f_{0}{}^{\prime }=\frac{-\Delta f\frac{2R_{q}}{c}+\left(\Delta f\frac{2mt_{s}}{c}+f_{0}\frac{2T_{r}}{c}\right)v_{0}}{T_{r}}\\ \mu =\frac{4\Delta f}{T_{r}c}v_{0}\\ T_{s}=nT_{r}\\ B{}^{\prime }=\mu NT_{r}=\frac{4N\Delta f}{c}v_{0} \end{array}\right.$$

Then, (Equation 7) can be regarded as a linear frequency modulation (LFM) signal with respect to $T_{s}$. Additionally, $f_{0}{}^{\prime }$ is center frequency, μ is the slope of the LFM, and $B{}^{\prime }$ is the bandwidth. Applying IDFT to (Equation 4) can be thought of as analogous to applying DFT to (Equation 7). In this LFM signal, the sampling period is $T_{r}$, so the frequency resolution is $\frac{T_{r}}{N}$. The spectrum of *r* will spread out around $f_{0}{}^{\prime }$. $f_{0}{}^{\prime }$ and $B{}^{\prime }$ are the functions of $v_{0}$. The shift of $f_{0}{}^{\prime }$ induced by $v_{0}$ is equal to the range shift caused by LPT in HRRP, and the spreading out of $B{}^{\prime }$ induced by $v_{0}$ corresponds to echo spread caused by QPT in HRRP. The influences of LPT and QPT can be eliminated by motion compensation. To ensure the quality of the moving target’s HRRP, the shift of $f_{0}{}^{\prime }$ and the spreading out of $B{}^{\prime }$ should be both less than 0.5 frequency resolution after motion compensation.
(9)$$\left\{\begin{array}{c} \frac{\left(\Delta f\frac{2mt_{s}}{c}+f_{0}\frac{2T_{r}}{c}\right)\Delta v_{L}}{T_{r}}\leq \frac{1}{2NT_{r}}\\ \frac{4N\Delta f}{c}\Delta v_{Q}\leq \frac{1}{2NT_{r}} \end{array}\right.$$
where $\Delta v_{L}$ and $\Delta v_{Q}$ represent compensation precision for LPT and QPT, respectively.
(10)$$\left\{\begin{array}{c} \Delta v_{L}\leq \frac{c}{4f_{0}NT_{r}}=\frac{c}{4f_{0}T_{N}}=\frac{\lambda }{4T_{N}}\\ \Delta v_{Q}\leq \frac{c}{8\Delta fN^{2}T_{r}}=\frac{c}{8BT_{N}} \end{array}\right.$$
where $T_{N}=NT_{r}$ is the period of the SF waveform. $\Delta v_{Q}$ is $\frac{f_{0}}{2B}$ times the value of $\Delta v_{L}$. In general, $f_{0}$ is much larger than *B*. So, the compensation precision for LPT is much higher than the compensation precision for QPT. $\Delta v_{L}$ is related to λ and $T_{N}$. Usually, the dimension of $T_{N}$ is milliseconds, and, for millimeter-wave radars, the dimension of $\Delta v_{L}$ may be meters per second, which means the SF waveform is highly sensitive to the target’s motion. The listed waveform parameter set 1 in Table 1 is substituted in (Equation 10), where $\Delta v_{L}$ and $\Delta v_{Q}$ are 0.558 m/s and 32.5521 m/s, respectively.

The methods based on DSF wavefrom and PCT can generate stationary HRRP of moving targets. It can simultaneously eliminate LPT and QPT without velocity estimation and compensation.

## 3. The Disadvantages of the Sequentially Transmitted DSF Wavform and the Existing PCT

The sequentially transmitted DSF waveform in [19] is shown in Figure 2b. Obviously, the sequentially transmitted DSF waveform contains two normal SF waveforms. The parameters of these two SF waveforms are the same except for the PRT. This signal can be mathematically expressed as:(11)$$\left\{\begin{array}{c} s_{1}{}^{\prime }\left(t\right)=\sum_{n=0}^{N-1}rect\left(\frac{t-nT_{r1}-\frac{T_{p}}{2}}{T_{p}}\right)e^{j(2\pi f_{n}t+\theta_{1_{n}})}\\ s_{2}{}^{\prime }\left(t\right)=\sum_{n=0}^{N-1}rect\left(\frac{t-nT_{r2}-NT_{r1}-\frac{T_{p}}{2}}{T_{p}}\right)e^{j(2\pi f_{n}t+\theta_{2_{n}})} \end{array}\right.$$

After frequency mixing and sampling, it is assumed that $T_{r2}=2T_{r1}=2T_{r}$. Then, the *n*th baseband signals of echo are derived as follows:(12)$$\left\{\begin{array}{c} r_{1}{}^{\prime }\left(n,m\right)=\sum_{q=0}^{Q-1}a_{q}e^{j(\varphi_{0}+\varphi_{1})}e^{j\{\varphi_{R_{q}}+\left[\varphi_{m}+\varphi_{L}\left(T_{r}\right)\right]+\varphi_{Q}\left(T_{r}\right)\}}\\ r_{2}{}^{\prime }\left(n,m\right)=\sum_{q=0}^{Q-1}a_{q}e^{j(\varphi_{0}+\varphi_{1}+\varphi_{2}{}^{\prime })}e^{j\{\varphi_{R_{q}}+\left[\varphi_{m}+\varphi_{T_{r}}{}^{\prime }+2\varphi_{L}\left(T_{r}\right)\right]+2\varphi_{Q}\left(T_{r}\right)\}} \end{array}\right.$$
where $\varphi_{2}{}^{\prime }=2\pi f_{0}\frac{2v_{0}NT_{r}}{c}$ and $\varphi_{T_{r}}{}^{\prime }=2\pi \Delta f\frac{2v_{0}NT_{r}}{c}n$. The existing PCT presented in [18,19] is expressed as:(13)$$r_{d}{}^{\prime }\left(n,m\right)=\frac{{\left[r_{1}{}^{\prime }\left(n,m\right)\right]}^{2}}{r_{2}{}^{\prime }\left(n,m\right)}$$

Substituting (Equation 12) into (Equation 13) and applying IDFT, the corresponding result is derived:(14)$$R_{d}{}^{\prime }\left(k,m\right)=\sum_{q=0}^{Q-1}a_{q}e^{j\left(\varphi_{0}+\varphi_{1}+\varphi_{k}{}^{\prime \prime }-\varphi_{2}{}^{\prime }\right)}\frac{sin[\pi \left(k-k_{q}+k_{v}{}^{\prime }\right)]}{Nsin[\frac{\pi }{N}\left(k-k_{q}+k_{v}{}^{\prime }\right)]}$$
where $\varphi_{k}{}^{\prime \prime }=\frac{N-1}{N}\pi \left(k-k_{q}+k_{v}{}^{\prime }\right)$ and $k_{v}{}^{\prime }=\frac{v_{0}mt_{s}}{\Delta r}-\frac{v_{0}NT_{r}}{\Delta r}$. In order to ensure the quality of HRRP, $|k_{v}{}^{\prime }|$ should be less than 0.5 [9]. The maximum speed of the target is computed as:(15)$$v_{max}{}^{\prime }=\frac{c}{4N^{2}\Delta fT_{r}}=\frac{c}{4BT_{N}}$$

If the velocity is less than $v_{max}{}^{\prime }$, the $k_{v}{}^{\prime }$ can be ignored. Then, (Equation 14) is similar to (Equation 5), which means the stationary HRRP of moving targes is obtained. Nevertheless, comparing (Equation 10) to (Equation 15), $v_{max}{}^{\prime }$ is only twice the maximum value of $\Delta v_{Q}$, which means sequentially transmitted DSF waveform is not suitable for high-speed targets.

The *n*th baseband signals of echo in (Equation 12) can be rewritten as:(16)$$\left\{\begin{array}{c} r_{1}{}^{\prime }\left(n,m\right)=A_{1}e^{j\phi_{1}}\\ r_{2}{}^{\prime }\left(n,m\right)=A_{2}e^{j\phi_{2}} \end{array}\right.$$

Adding the complex white Gaussian noise in the echo signal, the following equations are obtained:(17)$$\left\{\begin{array}{c} r_{1}{}^{\prime }\left(n,m\right)=A_{1}e^{j\phi_{1}}+n_{1}e^{j\varphi_{n_{1}}}=A_{1}{}^{\prime }e^{j\phi_{1}{}^{\prime }}\\ r_{2}{}^{\prime }\left(n,m\right)=A_{2}e^{j\phi_{2}}+n_{2}e^{j\varphi_{n_{2}}}=A_{2}{}^{\prime }e^{j\phi_{2}{}^{\prime }} \end{array}\right.$$
(18)$$\left\{\begin{array}{c} A_{1}{}^{\prime }=\sqrt{A_{1}^{2}+n_{1}^{2}+2A_{1}n_{1}cos\left(\phi_{1}-\varphi_{n_{1}}\right)}\\ A_{2}{}^{\prime }=\sqrt{A_{2}^{2}+n_{2}^{2}+2A_{2}n_{2}cos\left(\phi_{2}-\varphi_{n_{2}}\right)} \end{array}\right.$$
(19)$$\left\{\begin{array}{c} \phi_{1}{}^{\prime }=arctan\frac{A_{1}sin\phi_{1}+n_{1}sin\varphi_{n_{1}}}{A_{1}cos\phi_{1}+n_{1}cos\varphi_{n_{1}}}\\ \phi_{2}{}^{\prime }=arctan\frac{A_{2}sin\phi_{2}+n_{2}sin\varphi_{n_{2}}}{A_{2}cos\phi_{2}+n_{2}cos\varphi_{n_{2}}} \end{array}\right.$$
where $n_{1}e^{j\varphi_{n_{1}}}$ and $n_{2}e^{j\varphi_{n_{2}}}$ denote the complex white Gaussian noise samples, which are independent of each other. Substituting (Equation 17) into (Equation 13), the result is derived: (20)$$r_{d}{}^{\prime }\left(n,m\right)=\frac{{\left(A_{1}{}^{\prime }\right)}^{2}}{A_{2}{}^{\prime }}e^{j(2\phi_{1}{}^{\prime }-\phi_{2}{}^{\prime })}$$

As the two samples of complex white Gaussian noise are independent, $A_{1}{}^{\prime }$ is generally not equal to $A_{2}{}^{\prime }$. Moreover, when the SNR is reduced, $A_{2}$ and $\phi_{2}$ are approximately equal to $n_{2}$ and $\varphi_{n_{2}}+\pi$, respectively, with some probability. In this case, $A_{2}{}^{\prime }$ is approximately equal to zero, and the amplitude of $r_{d}{}^{\prime }\left(n,m\right)$ will be an extremum. These extremums generated by using the existing PCT are considered as impulse signals in the frequency domain and can be interpreted as DC signals in the time domain after applying IDFT. These DC signals generate false peaks after range profile splicing [20], which reduces the output SNR and influences target detection.

This paper presents a new method for generating stationary HRRP of moving targets. Applying cross-transmitted DSF waveform, the proposed method can adapt to high-speed targets. Additionally, the false peaks no longer appear by using the new PCT.

## 4. The Proposed Method for High-Speed Moving Targets

### 4.1. Signal Model of DSF Waveform

The designed cross-transmitted DSF waveform is shown in Figure 2c. This DSF waveform also contains two SF waveforms, but the pulses in these two SF waveforms are transmitted alternately. The transmitted signal of this cross-transmitted DSF waveform radar is mathematically expressed as:(21)$$\left\{\begin{array}{c} s_{1}\left(t\right)=\sum_{n=0}^{N-1}rect\left(\frac{t-nT_{r1}-\frac{T_{p}}{2}}{T_{p}}\right)e^{j(2\pi f_{n}t+\theta_{1_{n}})}\\ s_{2}\left(t\right)=\sum_{n=0}^{N-1}rect\left(\frac{t-nT_{r2}-\frac{T_{r1}}{2}-\frac{T_{p}}{2}}{T_{p}}\right)e^{j(2\pi f_{n}t+\theta_{2_{n}})} \end{array}\right.$$

In this paper, $T_{r2}$ is twice the value of $T_{r1}$. The *n*th baseband signals of echo can be expressed as follows:(22)$$\left\{\begin{array}{c} r_{1n}\left(t\right)=\sum_{q=0}^{Q-1}a_{q}rect\left(\frac{t-nT_{r1}-\frac{T_{p}}{2}-\tau_{1q}\left(t\right)}{T_{p}}\right)e^{-j2\pi f_{n}\tau_{1q}\left(t\right)}\\ r_{2n}\left(t\right)=\sum_{q=0}^{Q-1}a_{q}rect\left(\frac{t-nT_{r2}-\frac{T_{r}1}{2}-\frac{T_{p}}{2}-\tau_{2q}\left(t\right)}{T_{p}}\right)e^{-j2\pi f_{n}\tau_{2q}\left(t\right)} \end{array}\right.$$

Substituting (Equation 3) into (Equation 22), the following echo signals are derived as:(23)$$\left\{\begin{array}{c} r_{1}\left(n,m\right)=\sum_{q=0}^{Q-1}a_{q}e^{j(\varphi_{0}+\varphi_{1})}e^{j\{\varphi_{R_{q}}+\left[\varphi_{m}+\varphi_{L}\left(T_{r1}\right)\right]+\varphi_{Q}\left(T_{r1}\right)\}}\\ r_{2}\left(n,m\right)=\sum_{q=0}^{Q-1}a_{q}e^{j(\varphi_{0}+\varphi_{1}+\varphi_{2})}e^{j\{\varphi_{R_{q}}+\left[\varphi_{m}+\varphi_{T_{r}}+\varphi_{L}\left(T_{r2}\right)\right]+\varphi_{Q}\left(T_{r2}\right)\}} \end{array}\right.$$
where $\varphi_{2}\left(T_{r1}\right)=2\pi f_{0}\frac{T_{r1}}{c}v_{0}$, $\varphi_{T_{r}}=2\pi \Delta f\frac{v_{0}}{c}T_{r1}n$.

### 4.2. New Phase-Cancellation Technique

The new PCT is expressed as:(24)$$r_{d}\left(n,m\right)=\frac{r_{1}\left(n,m\right)}{{\left[r_{1}\left(n,m\right)\right]}^{*}}{\left[r_{2}\left(n,m\right)\right]}^{*}$$
where ${\left[\cdot \right]}^{*}$ denotes the conjugate operation. Assuming $T_{r2}=2T_{r1}=2T_{r}$, (Equation 23) can be rewritten as follows:(25)$$\left\{\begin{array}{c} r_{1}\left(n,m\right)=\sum_{q=0}^{Q-1}a_{q}e^{j(\varphi_{0}+\varphi_{1})}e^{j\{\varphi_{R_{q}}+\left[\varphi_{m}+\varphi_{L}\left(T_{r}\right)\right]+\varphi_{Q}\left(T_{r}\right)\}}\\ r_{2}\left(n,m\right)=\sum_{q=0}^{Q-1}a_{q}e^{j(\varphi_{0}+\varphi_{1}+\varphi_{2})}e^{j\{\varphi_{R_{q}}+\left[\varphi_{m}+\varphi_{T_{r}}+2\varphi_{L}\left(T_{r}\right)\right]+2\varphi_{Q}\left(T_{r}\right)\}} \end{array}\right.$$

By substituting (Equation 25) into (Equation 24), the following equation is produced as (for a detailed derivation of (Equation 26), please refer to Appendix A):(26)$$r_{d}\left(n,m\right)=\sum_{q=0}^{Q-1}a_{q}e^{j(\varphi_{0}+\varphi_{1}-\varphi_{2})}e^{j(\varphi_{R_{q}}+\varphi_{m}-\varphi_{T_{r}})}$$

Applying IDFT to (Equation 26), the next equation is generated.
(27)$$R_{d}\left(k,m\right)=\sum_{q=0}^{Q-1}a_{q}e^{j(\varphi_{0}+\varphi_{1}-\varphi_{2}+\varphi_{k}{}^{\prime })}\frac{sin[\pi \left(k-k_{q}+k_{v}\right)]}{Nsin[\frac{\pi }{N}\left(k-k_{q}+k_{v}\right)]}$$
where $\varphi_{k}{}^{\prime }=\frac{N-1}{N}\pi \left(k-k_{q}+k_{v}\right)$, $k_{q}=\frac{R_{q}}{\Delta r}$, $k_{v}=\frac{v_{0}mt_{s}}{\Delta r}-\frac{v_{0}T_{r}/2}{\Delta r}$. $\left(\varphi_{0}+\varphi_{1}-\varphi_{2}+\varphi_{k}{}^{\prime }\right)$ does not influence the amplitude of HRRP. $k_{v}$ represents the residual term of LPT. In this paper, it is supposed that the target does not move to the other sampling cells during the repetition interval of the DSF waveform. Thus, the range migration [16,21] can be ignored. The maximum speed of the target is calculated as follows:(28)$$v_{max}=\frac{t_{s}c/2}{2NT_{r}}$$

Now, substituting (Equation 28) in $k_{v}$, the result will be:(29)$$|k_{v}|\leq \frac{t_{s}\Delta f}{4}\leq \frac{T_{p}\Delta f}{4}$$

The design of the system parameters usually satisfies $T_{p}\Delta f\leq 1$ and $|k_{v}|\leq 0.25$, which means that the residual term can be ignored [9].

Finally, the stationary HRRP of the moving target is obtained as follows:(30)$$R_{d}\left(k,m\right)\approx \sum_{q=0}^{Q-1}a_{q}e^{j(\varphi_{0}+\varphi_{1}-\varphi_{2}+\varphi_{k}{}^{\prime })}sinc\left(k-k_{q}\right)$$

### 4.3. The Superiority of the Proposed Method

Compared to (Equation 15) to (Equation 28), the ratio of $v_{max}$ to $v_{max}{}^{\prime }$ is derived as follows:(31)$$\gamma =\frac{v_{max}}{v_{max}{}^{\prime }}=Nt_{s}\Delta f=\frac{B}{f_{s}}$$
where $f_{s}$ is the sampling frequency. In SF radars, *B* is much bigger than $f_{s}$. So, $v_{max}$ is $\frac{B}{f_{s}}$ times the value of $v_{max}{}^{\prime }$. The listed waveform parameter sets in Table 1 are substituted in (Equation 31), where the ratio is 15 and 85.3, respectively. This means that the cross-transmitted DSF waveform is more suitable for high-speed targets than the sequentially transmitted DSF waveform.

Similar to the process of (Equation 16) to (Equation 20), adding the complex white Gaussian noise in the echo signal and using the new PCT, the result is obtianed: (32)$$r_{d}\left(n,m\right)=\frac{r_{1}\left(n,m\right)}{{\left[r_{1}\left(n,m\right)\right]}^{*}}{\left[r_{2}\left(n,m\right)\right]}^{*}=A_{2}{}^{\prime }e^{j(2\phi_{1}{}^{\prime }-\phi_{2}{}^{\prime })}$$
(Equation 32) is different from (Equation 20), and $r_{d}$ do not generate extremums, so false peaks will no longer appear after applying IDFT to $r_{d}$.

## 5. Simulations

The simulations are performed to compare the quality of HRRP by using the nornal SF and the sequentially transmitted and cross-transmitted DSF waveforms, respectively. The parameter sets of DSF waveforms are listed in Table 1. A single-scattering-point target without noise is considered in simulations. The initial range of this target is 950Δr, and the values of velocity are 0 m/s, 60 m/s, and 900 m/s, respectively. Figure 3 shows the generated HRRPs by applying the parameter set 1. Figure 3a shows the HRRPs generated by using the normal SF. If the target is static, the HRRP of the target can be obtained accurately. However, the velocity of the moving target leads range shift and echo spread in HRRP. When the velocity is 60 m/s, the range shift is approximately 54, and it displays 10 after folding. When the velocity is 900 m/s, the echo spread in HRRP is serious. Figure 3b shows the generated HRRPs by using the sequentially transmitted waveform and the existing PCT, and when the velocity is less than $v_{max}{}^{\prime }$ ($v_{max}{}^{\prime }=65$ m/s), the existing method eliminates LPT and QPT effectively. In contrast, when the velocity is greater than $v_{max}{}^{\prime }$, the LPT still exists. Substituting v=900 m/s in $k_{v}{}^{\prime }$, $k_{v}{}^{\prime }$ would be 6.9, which is roughly equal to 7 and matches the range shift presented in Figure 3b. Figure 3c denotes the generated HRRPs by using the cross-transmitted DSF waveform and the new PCT, when the velocity is less than $v_{max}$ ($v_{max}=976$ m/s). In this case, the LPT and QPT are eliminated completely. The same phenomenon occurs by applying the parameter set 2, which is shown in Figure 4.

In the next step, the simulations are performed to compare the performance of the existing and proposed PCTs. To simplify the analysis, a single-scattering-point target with different input SNRs is used in simulations. The initial range of this target is 950Δr. As the velocity is 60 m/s, the LPT and QPT are eliminated by both PCTs. The performed simulations applying different parameter sets are shown in Figure 5. The existing PCT and the proposed PCT are represented by $PC_{1}$ and $PC_{2}$, respectively. As can be seen, there are many false peaks in the generated HRRP using $PC_{1}$, while only one true peak appears in the proposed method.

This paper gives a detailed analysis of Figure 5a. As mentioned in Section 3, the extremums are generated when the existing PCT is applied, as shown in Figure 6a. After performing IDFT, these extremums become DC signals, as presented in Figure 6b. It should be noted that the false peaks appear after the range profile splicing [20] using the data matrix presented in Figure 6b. Figure 6c presents the obtained generated data matrix by the proposed PCT from the echo data before applying IDFT, and the result after applying the IDFT is shown in Figure 6d. The moving target is presented in Figure 6d without false peaks.

Considering the different values of SNR, 100 simulations are performed based on $PC_{1}$ and $PC_{2}$ by using the parameter set 1. The SNRs of HRRP are compared in Figure 7. Figure 7a,b present the mean and variance of SNRs, respectively. As presented in Figure 7, the obtained mean of HRRP’s SNR using $PC_{2}$ is 15dB greater than the obtained SNR using $PC_{1}$. Moreover, the variance of HRRP’s SNR obtained using $PC_{2}$ is more stable than that obtained using $PC_{1}$.

In order to verify the performance of the proposed method in extended moving targets, an extended target possessing four scattering points is employed to perform the simulation. The initial ranges of these points are [950,952,955,960]Δr. Based on different parameter sets, the HRRPs corresponding to the stationary noiseless and noisy moving targets are shown in Figure 8. When the velocity is less than $v_{max}$ ($v_{max}$ is 976 m/s based on the parameter set 1 and $v_{max}$ is 8138 m/s based on the parameter set 2), the high-quality static HRRP of the extended moving target can be obtained, as shown in the mentioned figure.

## 6. Conclusions

The SF radar time-dividing transmits a sequence of single-frequency signals. The echo of the SF waveforms can be considered as samples in the frequency domain of the whole bandwidth. Applying IDFT to the echo can obtain the HRRP. The SF radar only needs to process a single-frequency signal at any time. Thus, the hardware of the system is easy to implement. Nevertheless, the SF waveforms are highly sensitive to the target’s motion. Thus, the velocity must be compensated for by the high-quality HRRP of moving targets.

This paper presents a new method for generating the stationary HRRP of moving targets. Using the new PCT, the proposed method can simultaneously eliminate LPT and QPT. Avoiding velocity estimation and compensation makes the processing easy. Compared to the existing method, the proposed PCT method can obtain higher-quality HRRP under low SNR conditions, and the cross-transmitted DSF waveform of the proposed method can be adapted to higher speed targets. The analysis and simulation results validate the proposed method. Moreover, the proposed method can be adopted for single-scattering-point and extended targets.

## Figures and Tables

**Figure 1 sensors-22-09191-f001:**
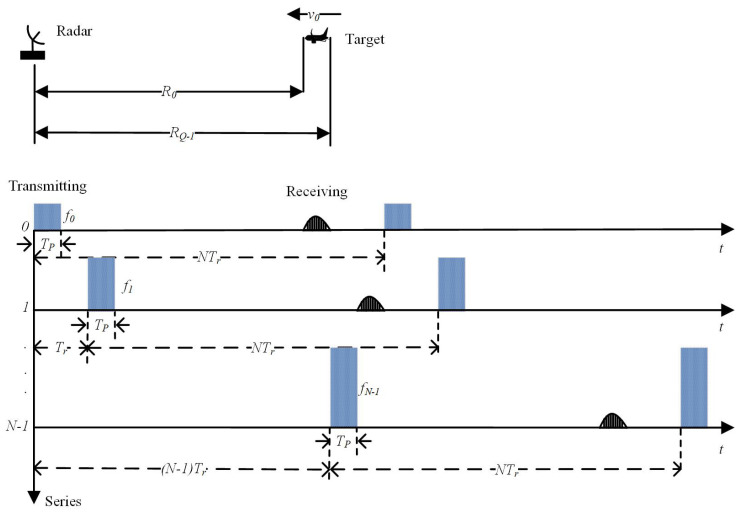
Scenario of a moving target in normal SF radar.

**Figure 2 sensors-22-09191-f002:**
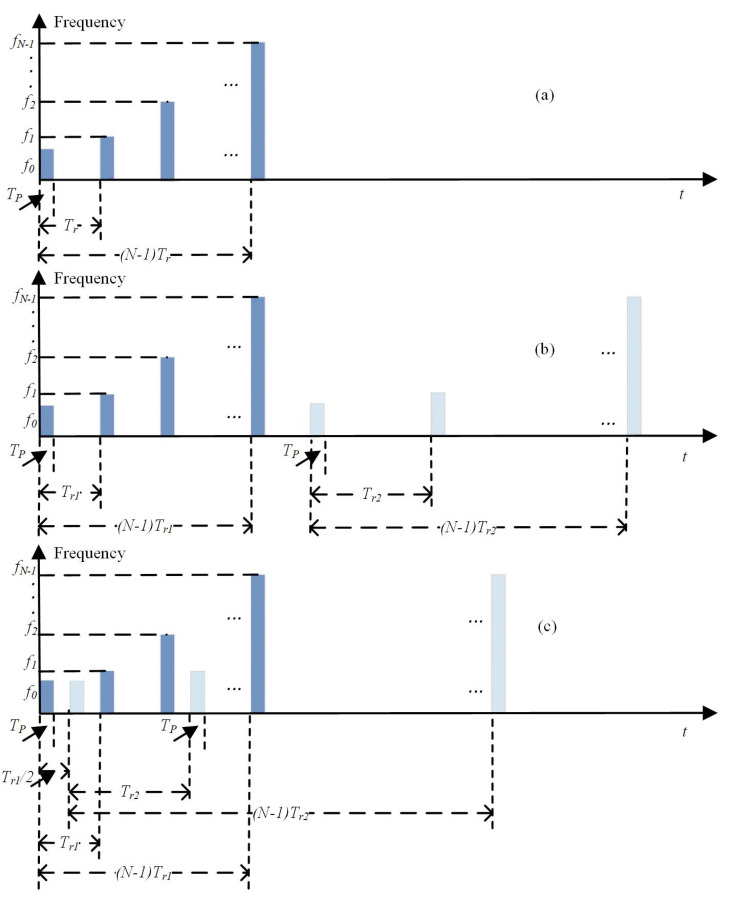
Normal SF, sequentially transmitted DSF, and cross-transmitted DSF waveforms. (**a**) Normal SF waveform. (**b**) Sequentially transmitted DSF waveform. (**c**) Cross-transmitted DSF waveform.

**Figure 3 sensors-22-09191-f003:**
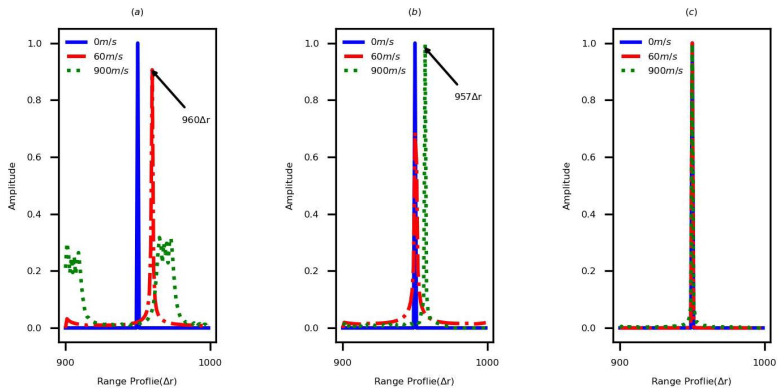
The HRRPs of the single-scattering-point target generated by the normal SF and the sequentially transmitted and cross-transmitted DSF waveforms in different velocities, based on the parameter set 1. (**a**) Normal SF. (**b**) Sequentially transmitted DSF waveform. (**c**) Cross-transmitted DSF waveform.

**Figure 4 sensors-22-09191-f004:**
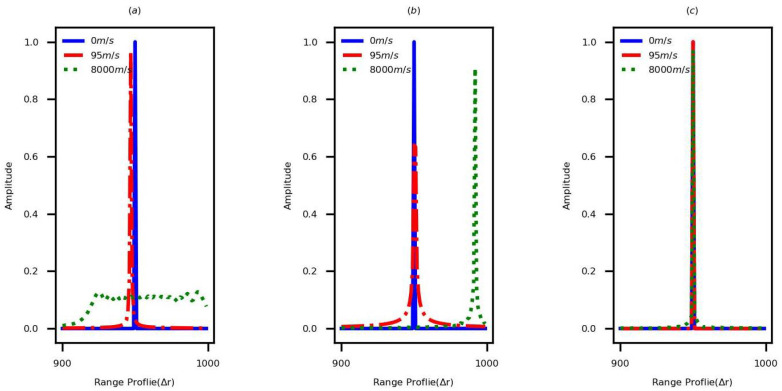
The HRRPs of the single-scattering-point target generated by the normal SF and the sequentially transmitted and cross-transmitted DSF waveforms in different velocities, based on the parameter set 2. (**a**) Normal SF. (**b**) Sequentially transmitted DSF waveform. (**c**) Cross-transmitted DSF waveform.

**Figure 5 sensors-22-09191-f005:**
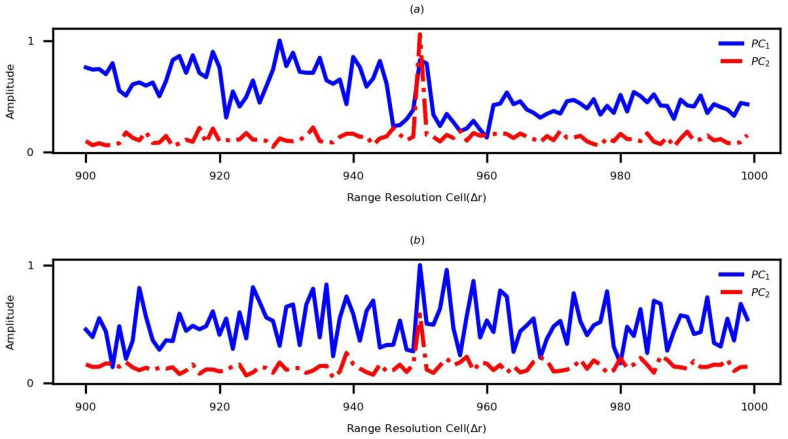
The HRRPs of the single-scattering-point target generated by the existing and the proposed PCTs based on different parameter sets. (**a**) Applying the parameter set 1. (**b**) Applying the parameter set 2.

**Figure 6 sensors-22-09191-f006:**
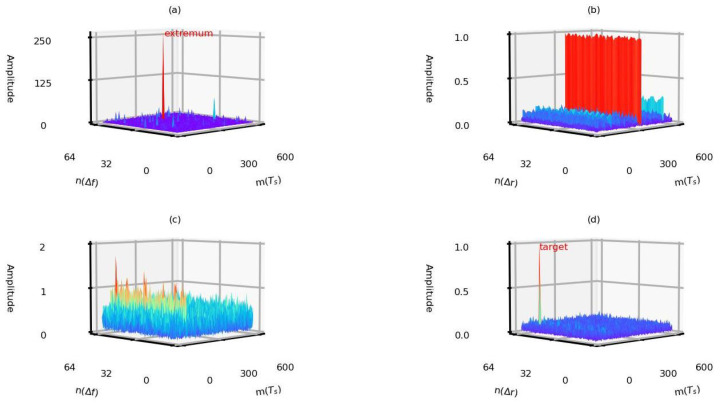
The generated data matrix by the existing method and the proposed method from the echo data based on parameter set 1. (**a**) The obtained data matrix using the existing PCT before applying IDFT. (**b**) The obtained data matrix using the existing PCT after applying IDFT. (**c**) The obtained data matrix using the proposed PCT before applying IDFT. (**d**) The obtained data matrix using the proposed PCT after applying IDFT.

**Figure 7 sensors-22-09191-f007:**
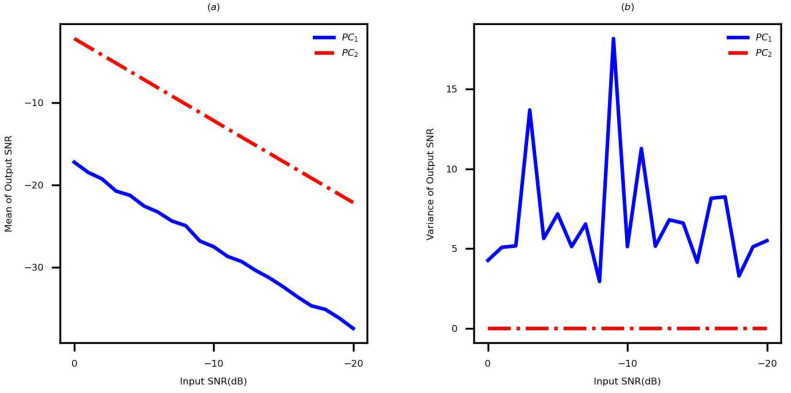
A comparison of the SNRs of HRRP generated by the existing method and the proposed methods. (**a**) The mean of HRRP’s SNR. (**b**) The variance of HRRP’s SNR.

**Figure 8 sensors-22-09191-f008:**
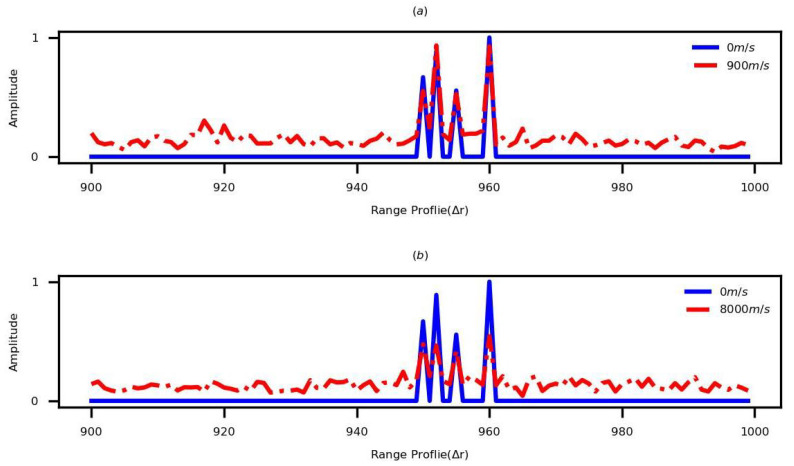
The HRRPs of the stationary noiseless target and the noisy moving target based on different parameter sets. (**a**) Applying the parameter set 1. (**b**) Applying the parameter set 2.

**Table 1 sensors-22-09191-t001:** The parameter sets of DSF waveforms.

	Parameter Set 1	Parameter Set 2
$f_{0}$	35 GHz	10 GHz
Δf	4.6875 MHz	8 MHz
*N*	64	256
$T_{p}$	0.2 us	0.125 us
$T_{r}$	60 us	1.5 us
$t_{s}$	$\frac{T_{p}}{4}$	$\frac{T_{p}}{3}$

## Data Availability

Not applicable.

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
