# Peer review of "A New Method for Moving-Target HRRP via Double Step Frequency Verified by Simulation"

_sensors, 2022, doi:10.3390/s22239191_

Round 1

Reviewer 1 Report

In order to address the problem of motion sensitivity in High Range Resolution Profile (HRRP) for moving target, the authors proposed a new method for signal process. Based on cross-transmitted Double-Stepped Frequency Waveform (DSFW) and Phase Cancellation Technique (PCT), the proposed method obtains stationary HRRP of the moving targets according to the Inverse Discrete Fourier Transform (IDFT) and complex multiplication. Through numerical simulation and comparison with traditional methods, the feasibility of the method is partially verified. But there are some defects.

1. In the research fields of High Range Resolution Profile (HRRP) radar, velocity sensitivity of moving targets is a common problem. Many attempts have been made to address this problem, and both the Double-Stepped Frequency Waveform (DSFW) and Phase Cancellation Technique (PCT) has been reported as mentioned in the References. However, the author does not explain the unique innovation of the method compared with the existing methods.    

2. Without any experiment, the author only uses numerical simulation under ideal conditions to verify the method.

3. The validation of the method should be verified by comparing it with the existing Double-Stepped Frequency Waveform (DSFW) and Phase Cancellation Technique (PCT) , rather than comparing it with the normal Stepped Frequency Waveform (SFW) without Phase Cancellation Technique (PCT).

4. It is better to simplify or delete the last part of the Chapter I and the first part of the Chapter II & III, and rewrite the manuscript according to the general style of academic papers.

5. In order to make the logical sequence of the manuscript more reasonable, it is better to change the sequence of Chapter II and Chapter III. In the Chapter II, it is better to make the deep analysis of moving target problem in normal Step-Frequency Waveform (SFW) and Phase Cancellation (PCT) firstly. In the Chapter III, it is better to put forward your new methods and carry out theoretical analysis according to the problems in the previous chapter.

6. All rigorous theoretical analysis is strongly dependent on the schematic diagram of the object. However, the only Figure 1 cannot clearly describe the relationship between waveform and radar system under different conditions.

Reviewer 2 Report

Globally, the manuscript is very well written and organized. The topic under study is relevant and deserves to be researched.

I recommend the authors to do a global final reading of the manuscript in order to correct some minor existing “bugs”; please refer to the attached commented document where some of these bugs are highlighted.

Additionally, I believe that the explanation following equation (1) will become clear if the values for “f_n” will be rewritten as “f_n = f0 + (n-1) \times \delta f” (or any other solutions that the authors consider better than this proposal).

Finally, in the paragraph following equation (5), the authors state “According to (5) and by ignoring \phi_m + \phi_Tr,…”. To what extent these values can be ignored? What will be reasonable bounds?

Round 2

Reviewer 1 Report

Major modifications must be made according to the comments
